# Characterization of blackbody inhomogeneity and its effect on the retrieval results of the GLORIA instrument

Anne Kleinert[1], Isabell Krisch[2], Jörn Ungermann[2], Albert Adibekyan[3], Berndt Gutschwager[3], and Christian Monte[3]

[1]Institut für Meteorologie und Klimaforschung (IMK-ASF), Karlsruher Institut für Technologie, Karlsruhe, Germany
[2]Institut für Energie- und Klimaforschung – Stratosphäre (IEK-7), Forschungszentrum Jülich GmbH, Jülich, Germany
[3]Physikalisch-Technische Bundesanstalt, Berlin, Germany

**Correspondence:** Anne Kleinert (anne.kleinert@kit.edu)

**Abstract.** Limb sounding instruments play an important role for the monitoring of climate trends, as they provide a good vertical resolution. Traceability to the SI via onboard reference or transfer standards is needed to compare trend estimates from multiple instruments. This study investigates the required uncertainty of these radiation standards to properly resolve decadal trends of climate relevant trace species like ozone, water vapor and temperature distribution for the Gimballed Limb Observer
for Radiance Imaging of the Atmosphere (GLORIA). Temperature nonuniformities of the onboard reference blackbodies, used for radiometric calibration, have an impact on the calibration uncertainty. The propagation of these nonuniformities through the retrieval is analyzed. A threshold for the maximum tolerable uncertainty of the blackbody temperature is derived, so that climate trends can be significantly identified with GLORIA.

## 1  Introduction

Remote sensing from satellites is crucial for a better monitoring of the Earth system, as only in this way a global data set covering also remote regions can be provided. Trace gas profiles with high vertical resolution are provided by limb sounders. Currently, limb satellites for observing the upper troposphere and stratosphere (UTS) are operating beyond design lifetime (e.g. SABER; Russell III et al., 1999 or ACE; Bernath et al., 2005) or already ceased their service (e.g. MIPAS; Fischer et al., 2008). No new limb satellite mission is approved, yet. Thus a gap of observations of the UTS with high altitude resolution
is foreseeable. This will lead to interruptions in atmospheric trend monitoring. The Gimballed Limb Observer for Radiance Imaging of the Atmosphere (GLORIA; Riese et al., 2014), an airborne Fourier Transform Spectrometer, can help to fill the gap. Furthermore, it makes a comparison of current satellite instruments with future ones possible, as it can provide reference measurements now and in future.

Several climate relevant species like ozone and water vapor, as well as the atmospheric temperature can be retrieved from
GLORIA measurements. To measure continuous climate trends with different instruments, the measurements have to be traceable to the SI via radiation standards. The present study aims to provide an uncertainty threshold for the radiation standards used for the calibration of the GLORIA instrument. This threshold value will be based on the requirement to resolve typical trends of atmospheric species like ozone, water vapor and temperature.

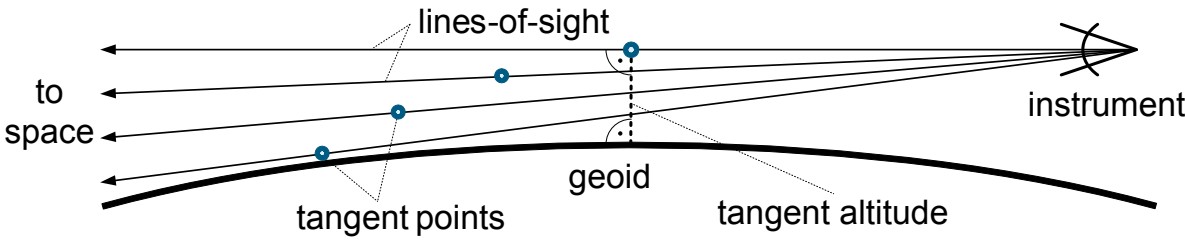

**Figure 1.** The viewing geometry of a limb sounder (Ungermann et al., 2013).

After an introduction to the GLORIA instrument and the radiometric calibration approach in Sect. 2, we investigate the impact of blackbody temperature uncertainties on retrieval results in Sect. 3, taking various types of blackbody nonuniformities into account. In Sect. 4, we present characterization results of the GLORIA calibration blackbodies in terms of absolute temperature uncertainties and nonuniformities. As an example, we apply these uncertainties to actual GLORIA measurements and derive the resulting retrieval error. All uncertainties given in the text shall be understood as $1\,\sigma$ values if not indicated otherwise.

## 2   The GLORIA instrument

### 2.1   Measurement principle

The Gimballed Limb Observer for Radiance Imaging of the Atmosphere (GLORIA) is an airborne Fourier Transform Spectrometer (FTS) for measuring atmospheric emission in the thermal infrared spectral range from 780 to $1400\,\mathrm{cm}^{-1}$. GLORIA combines a classical Michelson interferometer with a 2D detector array. The infrared radiation emitted by molecular vibrational and rotational bands along the line-of-sight (LOS) is measured (Fig. 1). The interferometer spectrally resolves this radiation to reveal characteristic emission and absorption bands.

GLORIA operates in two different modes, the dynamics mode and the chemistry mode. The chemistry mode has a high spectral sampling of $0.0625\,\mathrm{cm}^{-1}$, whereas the dynamics mode operates on a coarser spectral sampling of $0.625\,\mathrm{cm}^{-1}$. This coarser spectral sampling allows for faster interferogram acquisition and thus an improved spatial resolution along the flight track. From the intensities of the measured spectrally resolved radiation, atmospheric parameters like temperature and a variety of trace gases can be retrieved. A detailed description of the GLORIA measurement concept can be found in Friedl-Vallon et al. (2014).

The LOS of limb instruments is directed towards the limb of the Earth's atmosphere, hence, a good vertical resolution is achieved. With the 2D detector array $128 \times 128$ spectra can be measured simultaneously in the range from $5\,\mathrm{km}$ up to flight altitude. Typical images are narrowed to $48 \times 128$ (width $\times$ height) in favor of a shorter read out time. The horizontal imaging capability is currently not used for temperature and trace gas retrievals. Instead, the signal over each row is averaged to enhance the signal to noise ratio, and bad pixels are filtered prior to averaging. The vertical profiles are thus retrieved from a $1 \times 128$

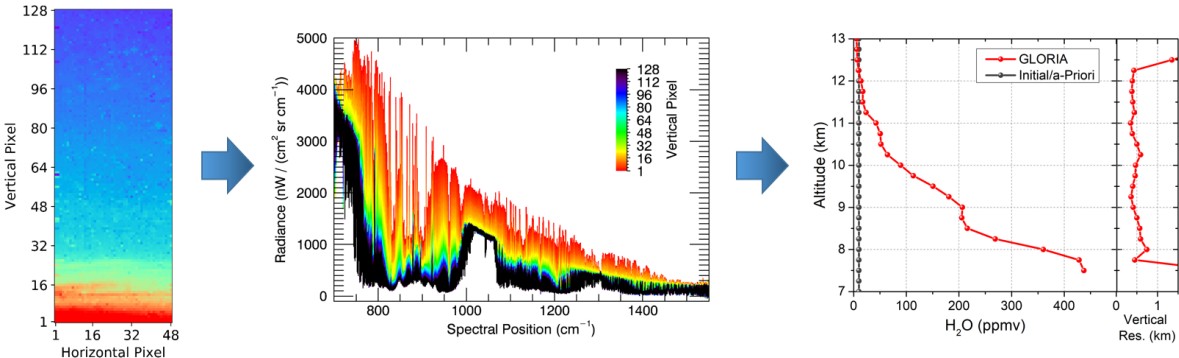

**Figure 2.** Brief illustration of the GLORIA data processing. Left: Raw single image of the detector array. Center: Set of 128 radiometrically and spectrally calibrated spectra representing one measurement. Right: Vertical profile of $H_2O$ (courtesy: W. Woiwode, KIT).

array. The standard viewing direction of GLORIA is $90°$ to the flight direction, such that 2D distributions of atmospheric parameters are measured along the flight path of the aircraft. Furthermore GLORIA is able to pan the LOS from $45°$ to $135°$ azimuth. This allows for tomographic retrievals in order to acquire 3D images of atmospheric features (Kaufmann et al., 2015; Krisch et al., 2017). The ability to pan the LOS is also used to point the FTS towards the two calibration blackbodies. The main

steps from raw measurements to vertical temperature and trace gas profiles are illustrated in Fig. 2.

## 2.2 Radiometric calibration

The GLORIA measurements in arbitrary intensity units have to be radiometrically calibrated to absolute radiances. Two on-board calibration sources are used to estimate a unique gain and offset for each detector pixel. The two-point calibration approach requires a linear detector response or a correction of any non-linear behavior of the detector system prior to radiometric

calibration.

The radiometric in-flight calibration uses two blackbodies at different temperatures (Olschewski et al., 2013). They are mounted outside of the spectrometer, such that the whole optical path including the entrance window is calibrated. In order to cover the FOV of the whole detector array, they have a rather large optical surface area of $126\,\text{mm} \times 126\,\text{mm}$. Since a revision of the blackbody design in 2015, the emitting back-surface of the blackbody consists of 25 by 25 pyramids of 8 mm height.

The backplane is wire eroded of one solid piece of aluminum for a good thermal conductivity. It features light traps at the bases of the pyramids to effectively avoid direct reflection of any incoming radiation and to increase the emissivity.

The pyramids as well as the casing are varnished with NEXTEL Velvet Coating 811-21. This design leads to an emissivity of better than 0.997 (see Sect. 4.1 below). Four Thermo-Electric Coolers (TECs), which provide the options of cooling and heating, are used to stabilize the temperature of each blackbody. The baseplate of the pyramid array is equipped with 10

Platinum Resistance Thermometers (PRTs) calibrated with a temperature uncertainty of $10\,\text{mK}$. The electronics measuring the resistance has an uncertainty of about 15 to $20\,\text{m}\Omega$, corresponding to a temperature uncertainty of about $50\,\text{mK}$. Four of these

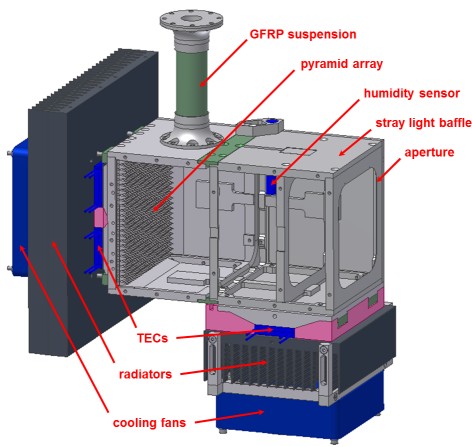

**Figure 3.** Design of the GLORIA calibration blackbodies (courtesy: F. Olschewski, BUW, Germany).

sensors are placed directly under the TECs and are used to control the heating or cooling rate. The other 6 sensors are used for monitoring. The temperature of the cold blackbody is stabilized at or slightly below the ambient temperature (typically around 230 to 250 K), the hot blackbody is heated to 30 to 40 K above the cold one. In order to avoid ice contamination and to suppress straylight, a baffle is mounted in front of the casing of each blackbody. The baffle is cooled with two TECs to

2 K below the temperature of the pyramid array. The blackbodies are insulated with polystyrene foam sheets and the baffle system is thermally decoupled from the pyramid section with a layer of Glass-Fiber Reinforced Plastic (GFRP). A sketch of the blackbodies is given in Fig. 3

The gain and offset of an instrument are defined as the multiplicative (gain, $g$) and additive (offset, $L_0$) factors, which must

be applied to the true radiation $L$, in order to get the signal measured by the instrument $S$ (see e.g. Revercomb et al., 1988):

$$S = g(L + L_0) \tag{1}$$

These factors differ for each pixel and wavelength. Gain and offset can be determined from measurements of two blackbody sources with known temperature. Assuming an emissivity of 1, the gain is calculated as

$$g = \frac{S_{\mathrm{bb2}} - S_{\mathrm{bb1}}}{B(T_{\mathrm{bb2}}) - B(T_{\mathrm{bb1}})} \tag{2}$$

and the offset as

$$L_0 = \frac{S_{\mathrm{bb1}}}{g} - B(T_{\mathrm{bb1}}), \tag{3}$$

with $B(T)$ being the Planck function of the given temperature. The effect of a non-perfect emissivity is discussed in Sect. 4. More details on the calibration concept of GLORIA are given by Kleinert et al. (2014) and Olschewski et al. (2013).

The calibrated spectral radiation $L$ can be calculated from the measurements by dividing the measured radiation $S$ by the gain $g$ and subtracting the offset $L_0$.

$$L = \frac{S}{g} - L_0, \tag{4}$$

## 2.3 Impact of blackbody temperature uncertainty on radiances

The surface area of the backplane of the GLORIA blackbodies with a size of $126\,\mathrm{mm} \times 126\,\mathrm{mm}$ is rather large. It is a technically challenging task to get a completely uniform temperature distribution over this big area. In reality, the temperature varies across the blackbody, due to thermal gradients inside the backplane, the air flow around the pyramids and an imperfect temperature regulation.

   If the true blackbody temperature $T_{\mathrm{bb}}^t = T_{\mathrm{bb}} - \Delta T_{\mathrm{bb}}$ deviates from $T_{\mathrm{bb}}$, which is used for calculating $g$ and $L_0$, the true

gain $g^t$ and true offset $L_0^t$ differ from $g$ and $L_0$ as follows:

$$g^t = \frac{S_{\mathrm{bb2}} - S_{\mathrm{bb1}}}{B(T_{\mathrm{bb2}}^t) - B(T_{\mathrm{bb1}}^t)} = g\alpha \tag{5}$$

with

$$\alpha = \frac{B(T_{\mathrm{bb2}}) - B(T_{\mathrm{bb1}})}{B(T_{\mathrm{bb2}}^t) - B(T_{\mathrm{bb1}}^t)} \tag{6}$$

and

$$L_0^t = \frac{S_{\mathrm{bb1}}}{g^t} - B(T_{\mathrm{bb1}}^t). \tag{7}$$

   This leads to a deviation $\Delta L$ of the originally calculated calibrated radiation $L$ from the true calibrated radiation $L^t$. The calibrated spectra can be expressed in terms of the "true" calibrated spectra $L^t = S/g^t - L_0^t$:

$$
\begin{aligned}
L &= \frac{S}{g} - L_0 & (8)\\[4pt]
&= \frac{S - S_{\mathrm{bb1}}}{g^t}\alpha + B(T_{\mathrm{bb1}}) & (9)\\[4pt]
&= \left( \frac{S - S_{\mathrm{bb1}}}{g^t} + B(T_{\mathrm{bb1}}^t) - B(T_{\mathrm{bb1}}^t) \right)\alpha + B(T_{\mathrm{bb1}}) & (10)\\[4pt]
&= (L^t - B(T_{\mathrm{bb1}}^t))\alpha + B(T_{\mathrm{bb1}}). & (11)
\end{aligned}
$$

   Thus, the deviation $\Delta L$ is

$$
\begin{aligned}
\Delta L &= L - L^t & (12)\\[4pt]
&= L^t(\alpha - 1) - B(T_{\mathrm{bb1}}^t)\alpha + B(T_{\mathrm{bb1}}). & (13)
\end{aligned}
$$

This deviation is a combination of a scaling compared to the true calibrated measurements and an additive offset.

   If one blackbody measurement is replaced by a deep space measurement, the deviation simplifies to a simple scaling of the true calibrated measurements:

$$\Delta L^{\mathrm{bb,ds}} = L^t \left( \frac{B(T_{\mathrm{bb}})}{B(T_{\mathrm{bb}}^t)} - 1 \right), \tag{14}$$

as the emission of deep space can be assumed to be effectively zero.

## 2.4 Other sources of uncertainty

Uncertainties in the gain function do not only come from uncertainties of the temperature and emissivity of the blackbody but also from uncertainties in the determination of the detector non-linearity and the instrument self-emission (offset). These uncertainties are currently estimated to about 2 - 3 % (k=2), with k being the uncertainty coverage factor (see, e.g., https://physics.nist.gov/cuu/Uncertainty/coverage.html). Further characterization is under way in order to reduce these uncertainties in future processing versions.

Beside the uncertainties in the calibrated spectra, the quality of the retrieval results also relies on the quality of the spectroscopic data and the impact of interfering species in the selected microwindows. Spectroscopic uncertainties are, however, systematic over time and cancel out in case of trend analyses. Furthermore, the quality of the spectroscopic data may improve in the future. Some interfering species impose uncertainties within the troposphere in the same order as the blackbody inhomogeniety. Further improvements on the retrieval are likely able to reduce these.

With the current data processing and retrieval setup, the blackbody temperature uncertainties are estimated to be one of the leading sources of uncertainty, together with other uncertainties in the gain determination and the impact of interfering species. Since the other major sources of uncertainty are expected to be reduced with better instrument characterization and an optimized retrieval setup, the focus of this study is on the impact of the blackbody temperature uncertainties on the retrieval results relevant for the detection of climate relevant trends.

## 3 Impact of blackbody temperature uncertainties on retrieval results

### 3.1 Study Concept

A Monte Carlo approach (e.g. Metropolis and Ulam, 1949; Press, 1968) is used for this part of the study. Therefore, many random temperature fluctuations of the blackbodies are generated and the effects of these fluctuations on the retrieved climate variables are calculated. Afterwards, statistical information on the perturbed retrieval results is derived. Only vertical temperature variations are taken into account, as the row averaged radiation is used for the standard GLORIA retrievals (see Sect. 2.1). Consequently, a random temperature deviation is created for each vertical pixel around the measured mean blackbody temperature $T_{\mathrm{bb}}$. These deviations $\Delta T_{\mathrm{bb}}$ are defined over the detector through the covariance matrix $\mathbf{COV}$ with standard deviation $\sigma$ and correlation length $c$:

$$\mathbf{COV}(i,j) = \sigma^2 \cdot e^{-\frac{1}{2} \cdot \left(\frac{i-j}{c}\right)^2} \tag{15}$$

with $i$ and $j$ denoting the 128 averaged rows of the detector. Examples for these temperature deviations $\Delta T_{\mathrm{bb}}$ are shown in Fig. 4 for different correlation lengths. A correlation length of 0 means completely uncorrelated deviations and infinity completely correlated (i.e., constant) ones. Due to the optical setup and the thermal conductivity of the blackbody surface, typical correlation lengths of the GLORIA blackbodies are expected to be in the range of 10 to 100 pixels. Temperature variations across the emitting surface have to be rather smooth because of the thermal conductivity of the material. Furthermore, the

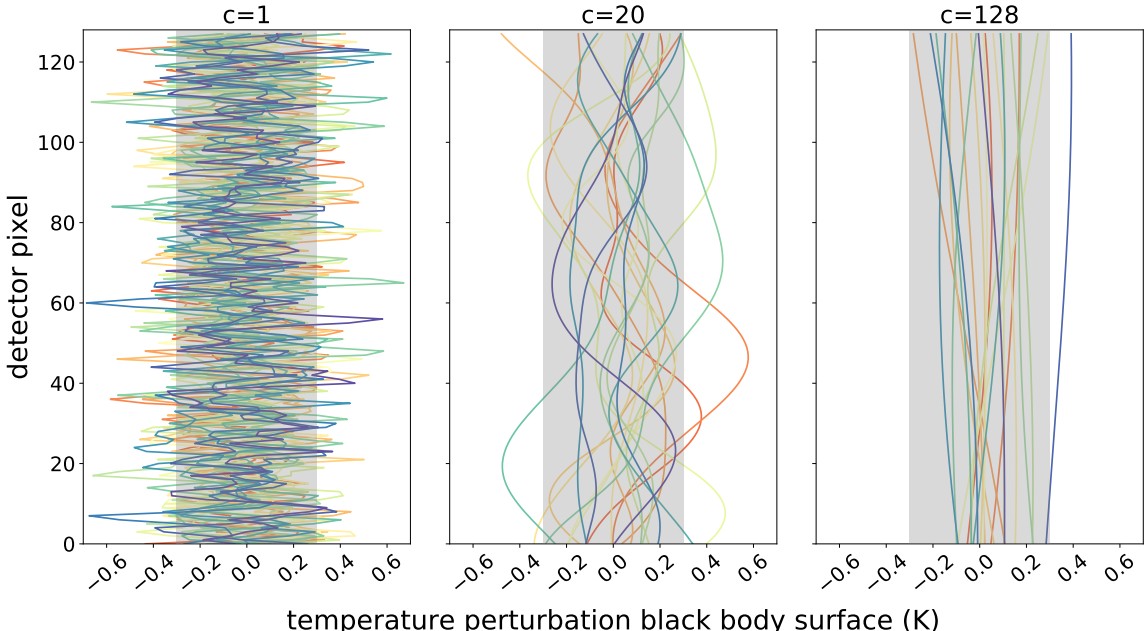

**Figure 4.** Random temperature deviations for the 128 vertical detector pixels with different correlation lengths $c$. The temperature deviations in the left panel have a correlation length of one vertical pixel, meaning that they are nearly uncorrelated. In contrast the temperature deviations in the right panel are nearly fully correlated over the whole detector (correlation length of 128 pixels). In the middle a correlation length of 20 pixels is depicted. All variations have a standard deviation of $\sigma = 0.2$. The gray shade marks the $3\sigma$ area.

instrument is focused to infinity and the picture of the blackbody backplane on the detector is not sharp. Therefore correlation lengths below 10 pixels are not expected.

256 random temperature distributions are generated for each pair of standard deviation and correlation length. These blackbody temperature deviations transform into radiation deviation distributions $\Delta \boldsymbol{L}$ according to Eq. (14).

This study uses atmospheric profiles from a climatology (Remedios et al., 2007) in the mid-latitudes to simulate infrared spectra as they would be measured by GLORIA at 15 km flight altitude. The GLORIA Measurement Simulator uses vertical profiles of relevant trace species and of pressure and temperature from the climatology and combines them in a forward model with the GLORIA measurement geometry. This results in one infrared spectrum for each of the 128 vertical pixels. Typical noise values of the GLORIA instrument are added to these spectra. These synthetic measurements are transferred

back into to the geophysical quantities of temperature and trace gas volume mixing ratios according to the retrieval setup described in Ungermann et al. (2015) for the GLORIA dynamics mode. A gain matrix $\mathbf{G}$ is calculated in this retrieval process, which correlates measurement deviations ($\Delta \boldsymbol{L}$) with retrieval result uncertainties ($\Delta \boldsymbol{x}$). As the radiation errors caused by the blackbody inhomogeneities are small compared to the simulated radiations, the retrieval can be linearized and the retrieval gain matrix $\mathbf{G}$ can be assumed to be constant for the small deviations $\Delta \boldsymbol{L}$ around the original radiation (Rodgers, 2000). Therefore,

the influence of the blackbody temperature deviations on the retrieval result of atmospheric trace gases and temperature can be

**Table 1.** Long term trends in temperature, ozone, and water vapor in the UTLS region and the derived target and threshold values for GLORIA retrievals.

|  | temperature | ozone | water vapor |
|---|---|---|---|
| 10 year trend | 1 K | 5% | 0.5 ppmv |
| reference | Schmidt et al. (2010) | WMO (2014) | Solomon et al. (2010) |
| target (threshold) | 0.08% (0.16%) | 1% (2%) | 2.5% (5%) |

calculated as follows:

$$\mathbf{G} \cdot \Delta \boldsymbol{L} = \Delta \boldsymbol{x}. \tag{16}$$

The retrieval uncertainty distribution is estimated through a fit of a Gaussian shaped probability distribution to the 256 $\Delta \boldsymbol{x}$ vectors of each pair of correlation length and standard deviation of the original $\Delta \boldsymbol{T}_{\mathrm{bb}}$ distribution. This allows to estimate a standard deviation and a correlation length of the retrieval uncertainty probability distribution. The results are presented in Sect. 3.2.

## 3.2   Study Results and Discussion

The measurement uncertainty of GLORIA should ideally be small compared to the decadal trends. We therefore choose a target and a threshold value for the accuracy of 20 % and 40 % of the decadal trends, respectively. Table 1 shows the decadal trends in temperature, ozone, and water vapor together with the resulting target and threshold values for these entities.

As described in Sect. 3.1, a Monte Carlo approach is used to estimate the propagation of temperature nonuniformities of the blackbody surface through the retrieval. Fig. 5 shows the results of this study. The standard deviations and correlation lengths of the retrieval errors are displayed for different standard deviations and correlation lengths of the random deviations of the blackbody temperature. These retrieval errors are averaged over the altitude range 7 - 14 km.

As expected, the correlation length of the retrieval error increases with the correlation length of the input for all species (see right panels in Fig. 5). For all species shown, the correlation lengths of the retrieval errors are around 1 - 1.5 km for blackbody nonuniformities correlated over less than 50 pixels. For temperature deviations over larger areas, the correlation lengths in the retrieval errors grow accordingly. These more extended errors introduce a bias in the measurements but do not influence the general structures of the measured species. For some scientific use cases, like for example gravity wave research, this is more acceptable than errors with short correlation lengths.

The standard deviations of the results depend strongly on the retrieved species. For typical correlation lengths of the GLORIA blackbody temperature deviations (10 to 100 pixels) the accepted threshold of 2 % error for the ozone retrieval is reached for standard deviations of 0.2 - 2 K. Therefore, a required threshold of 200 mK and a target of 100 mK for the temperature standard deviation of the blackbody are defined through the ozone retrieval. These values can be relaxed for temperature deviations with

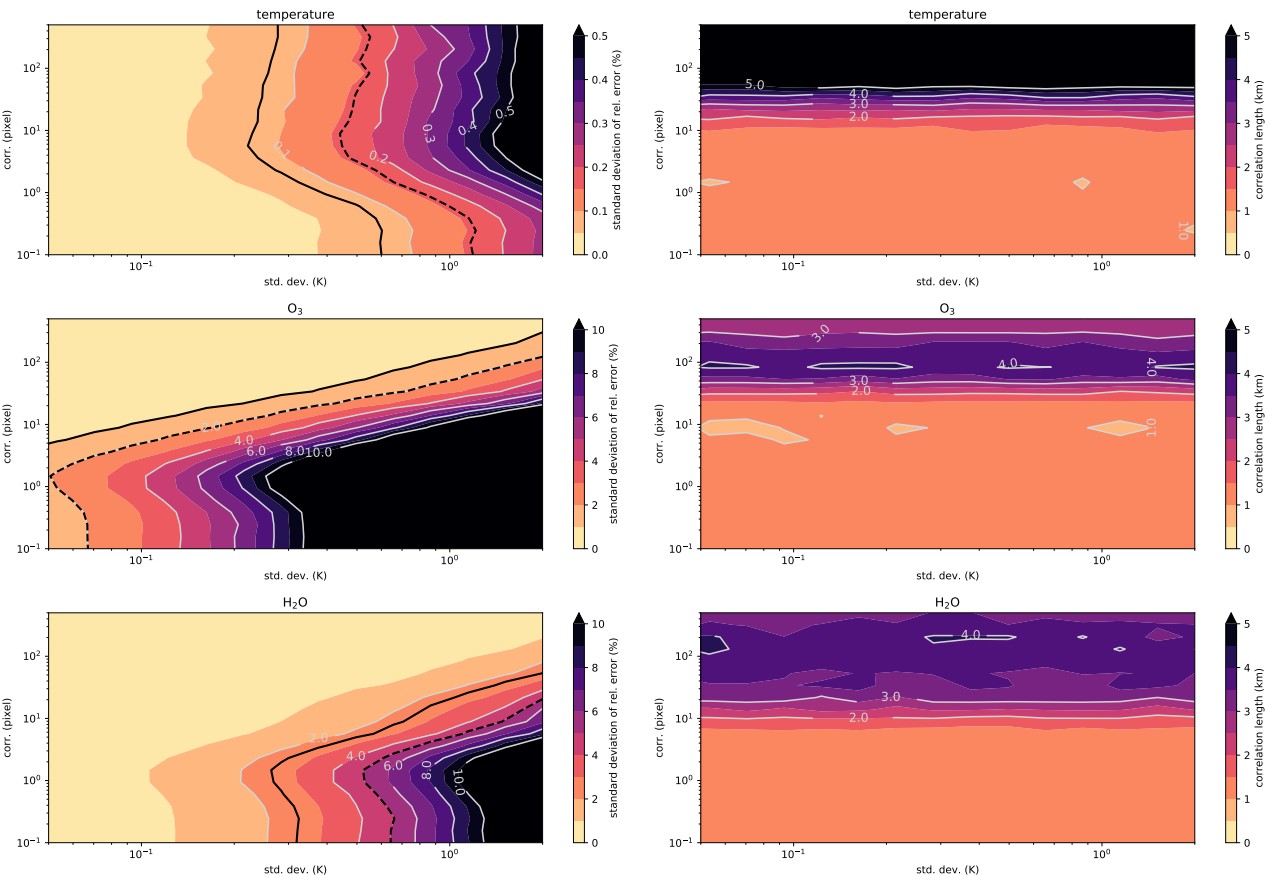

**Figure 5.** Results of the Monte Carlo approach. Plotted are the standard deviations (left panels) and correlations lengths (right panels) of the retrieval error distributions. The x-axis shows the standard deviation of the original random variation of the blackbody temperature, the y-axis the according correlation length. The top row represents the temperature retrieval, in the middle ozone is shown, and the bottom depicts water vapor. The black solid (dashed) line in the left plots indicates the error target (threshold) due to the long term trend of the respective atmospheric species.

longer correlation lengths, as can be seen in the plot in the middle left of Fig. 5. For the temperature retrieval the threshold of 0.16 % uncertainty is reached for all correlation lengths above two pixels at around 400 mK. The target uncertainty is reached already at 200 mK. For water vapor retrievals the threshold and target values are located at a standard deviation of 500 mK and 250 mK for correlation lengths of one pixel but can be relaxed for more correlated nonuniformities.

Thus, for the GLORIA instrument and the applied retrieval setup, the ozone retrieval sets the threshold requirement of the standard deviation of the blackbody uncertainties to 200 mK and the desired target to 100 mK.

## 4    Blackbody Characterization

The absolute radiance temperature of the GLORIA onboard reference blackbodies is regularly calibrated at the Physikalisch-Technische Bundesanstalt (PTB), the national metrology institute of Germany. A rigid and thorough calibration procedure
has been established to characterize and calibrate the blackbodies traceable to the national standards of temperature and the international temperature scale ITS-90. The surface radiation temperature is directly measured and compared to blackbody radiation from high-quality cavity radiators linked to the primary standards of temperature radiation (Monte et al., 2014). Additionally, the blackbodies are spectrally characterized.

For this purpose, the blackbodies are mounted in the source chamber of the reduced background calibration facility (RBCF).
The RBCF serves as facility for the measurement of spectral radiance and radiation temperature of radiation sources and the signal temperature characteristic of radiation thermometers under reduced infrared background radiation in a temperature range from -170 °C to 430 °C traceable to the ITS-90. Its major vacuum components are a source chamber, accommodating the blackbody reference radiators; an optomechanical unit, allowing folding mirrors and an optical chopper to be inserted into the line of sight; a beam line; and a detector chamber, which houses the Vacuum Infrared Standard Radiation Thermometer
(VIRST). Additionally, an off-axis ellipsoidal mirror in the detector chamber allows the imaging of the radiation from the source chamber into a vacuum FTS for spectrally resolved measurements.

The blackbody surface radiation temperature is measured with the Vacuum Infrared Standard Radiation Thermometer (VIRST). VIRST is a stand-alone transfer radiation thermometer which features a vacuum-insulated and thermally stabilized housing, a selected thermopile detector, dedicated readout electronics and an aspheric germanium lens. A band-pass filter in
front of the detector limits the sensitive spectral range from 8 μm to 14 μm. VIRST achieves a temperature resolution or Noise-Equivalent Temperature Difference (NETD) of better than 10 mK at 0 °C. Using VIRST as transfer radiation thermometer, the GLORIA blackbodies are calibrated against the Vacuum Low-Temperature BlackBody (VLTBB), operable in a temperature range from -173 °C to +177 °C. The calibration range is from -40 °C to +40 °C. The typical uncertainty for the GLORIA blackbody surface radiation temperature at -40 °C at a wavelength of 10 μm is 100 mK (k=2) and is decreasing for higher
temperatures.

The measurement spot of VIRST has a diameter of about 10 mm. It is placed directly over the positions of the 10 temperature sensors of the blackbodies, allowing a direct comparison of the radiation surface temperature with the temperature measured at the PT100 sensors. Measurements are taken at temperatures ranging from -40 °C to +40 °C in 10 K steps. Furthermore,

surface scans are performed at temperatures of -30 °C, 0 °C, and 20 °C. For these measurements, the complete surface of the blackbody is scanned in steps of 4.5 mm.

The spectral characterization is done by measuring the spectral radiance at three locations with the vacuum Fourier-transform spectrometer of the RBCF with respect to the reference blackbody VLTBB in the spectral range from 6.3 μm to 20 μm at temperatures of -30 °C, 0 °C and 20 °C.

## 4.1 Effective emissivity of the blackbodies

On the basis of the determined spectral radiances, a lower threshold value for the effective emissivity of the blackbodies can be calculated. Since no significant spectral features are found, the blackbodies can be considered to be gray bodies. Assuming an effective emissivity $\epsilon_{\text{eff}}$, the measured radiance of the blackbodies $L_{\text{meas}}$ is given as

$$L_{\text{meas}} = \epsilon_{\text{eff}} B(T_{\text{BB}}) + (1 - \epsilon_{\text{eff}}) B(T_{\text{amb}}) \tag{17}$$

with $B(T_{\text{BB}})$ and $B(T_{\text{amb}})$ being the Planck function of the blackbody and ambient temperature, respectively.

The radiance of an ideal blackbody $B(T_{\text{BB}})$ can thus be calculated as:

$$B(T_{\text{BB}}) = \frac{L_{\text{meas}}(T) - (1 - \epsilon_{\text{eff}}) B(T_{\text{amb}})}{\epsilon_{\text{eff}}} \tag{18}$$

This spectral radiance can be transformed in a radiance temperature by inverting Planck's law at every wavelength and yields a spectral distribution of radiance temperatures. This was done for the three effective emissivities 1.000, 0.997 and 0.995 and the results are shown in Fig. 6. The calculated values are given by the individually colored dots in the figure. For each assumed effective emissivity, a linear fit was performed in order to identify possible trends. The calculated radiance temperatures for the lower emissivity of 0.995 show a significant slope towards shorter wavelengths. This slope can only be real if either the wall coating has a significant slope or if the temperature non-uniformity within the measured area is of a similar magnitude as the slope, i.e. 200 mK. The coating Nextel 811-21 can be safely assumed to have a constant value of 0.97 in the spectral range from $500 \, \text{cm}^{-1}$ to $1500 \, \text{cm}^{-1}$ with only a slight spectral variability within a given uncertainty of about 0.01 but no significant overall spectral trend or slope (Adibekyan et al., 2017).

In order to exclude a temperature non-uniformity in the order of 200 mK, lateral scans were performed with VIRST over the field-of-view of the spectrometer on a 6 x 6 grid with 4 mm step size. These measurements show a uniformity of the radiance temperature of better than 18 mK (peak to peak). Therefore, we conclude that the assumed effective emissivity of 0.995 is too low. The slope of the radiance temperatures for the assumed effective emissivity of 0.997 is smaller but still significant. However, considering the uncertainties of the measurement and the VLTBB reference blackbody which are around 100 mK, we assume 0.997 as a lower limit for the effective emissivity of the GLORA reference blackbodies. A non-perfect emissivity leads to deviations between the blackbody surface radiation temperature measured by VIRST and the temperature measured by the PT100 sensors if the ambient temperature differs from the one of the blackbodies. The effects of a non-perfect emissivity are therefore included in the temperature characterization measurements and handled as temperature uncertainties.

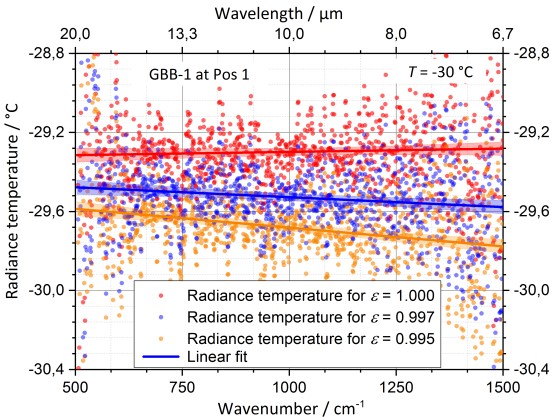

**Figure 6.** Spectral distribution of the radiance temperatures of the GLORIA reference blackbody BB1 for varying effective emissivities.

## 4.2 Temperature characterization results

The measurement results for the different temperatures are shown in Fig. 7 for one of the blackbodies (BB1). The temperature measured by the PT100 sensors generally agrees with the temperature measured by VIRST within $\pm\,100\,\mathrm{mK}$. The measured radiation temperatures are generally higher than the measured contact temperatures when operating the blackbody below the ambient temperature and vice versa when operating it above ambient temperature. This can be explained by the finite thermal conductivity of the aluminum and the Nextel coating, as well as a non-perfect emissivity. Furthermore there is a spread of about $30\,\mathrm{mK}$ between the 10 temperature sensors.

Two surface scans are shown in Figs. 8 and 9 on the left for BB1 at -30 °C and BB2 at +20 °C, respectively. The temperature uniformity is within the required threshold of $100\,\mathrm{mK}$, but BB1 shows a systematic deviation in the lower right area of the surface. BB2 at +20 °C is very uniform and dominated by noise. These two surface scans are used to estimate a realistic blackbody uncertainty including its distribution over the optical surface.

The area covered by the detector array is approximately from -15 mm to +15 mm horizontally and from -40 mm to +40 mm vertically. The detector lens of GLORIA is focused at infinity in order to provide a sharp picture of the horizon. Since the blackbodies are placed closely to the spectrometer, the optical surface is not sharply imaged onto the detector. One detector pixel 'sees' a spot of about 30 mm diameter on the blackbody surface. In order to quantify the blackbody temperature uncertainty relevant for each pixel, the values shown in Figs. 8 and 9 on the left have been integrated over 30 mm spots around the center position of each detector pixel image. The resulting temperature uncertainty for each pixel is shown in the right panel of the figures. This uncertainty has been used to calculate $T_{\mathrm{bb}}$ and $T_{\mathrm{bb}}^{t}$ for each pixel and with this the radiance uncertainty for each pixel using Eq. 13.

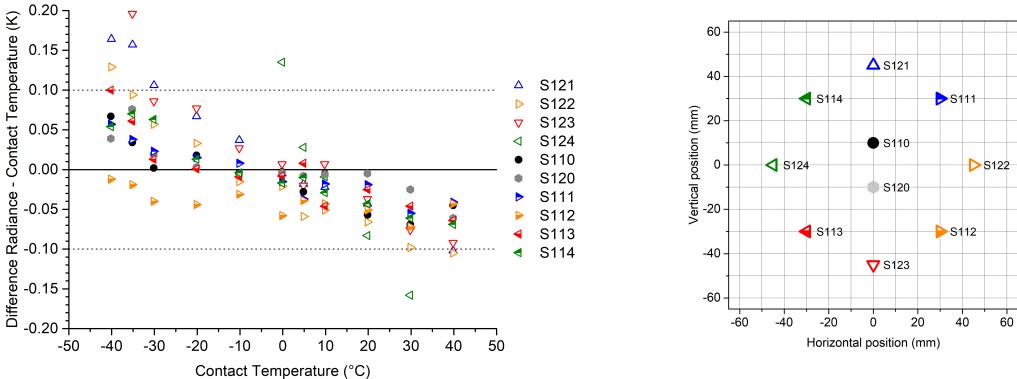

**Figure 7.** Differences between surface radiance temperatures and temperatures measured with the PT100 sensors (contact temperature) for BB1 at different temperatures. The approximate positions of the sensors under the optical surface are shown on the right.

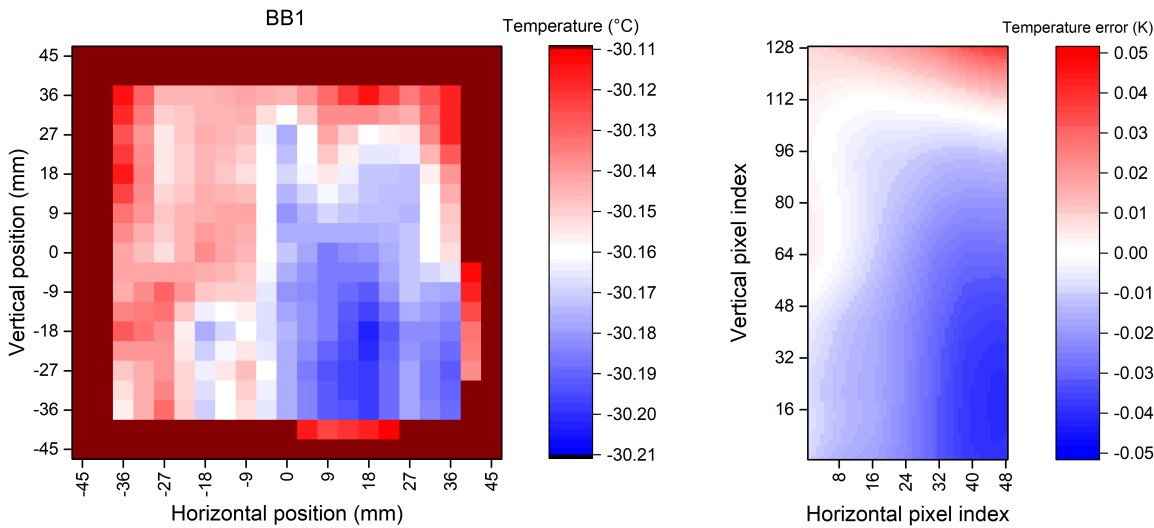

**Figure 8.** Measured surface radiance temperatures at -30.15 °C (243 K) for BB1 (left) and estimated resulting temperature error for each of the 48 x 128 detector pixels (right)

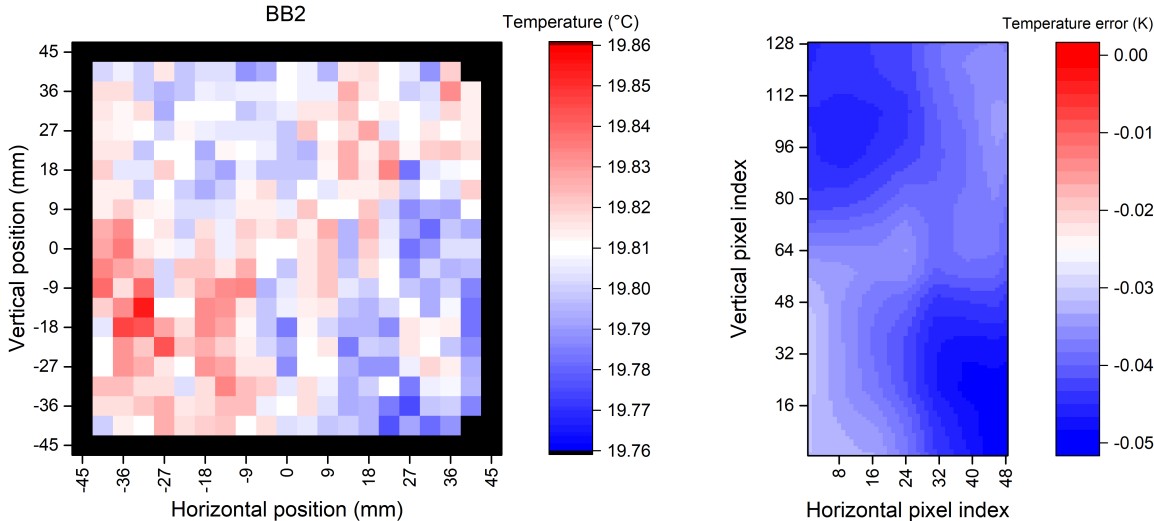

**Figure 9.** Same as Fig. 9 but for BB2 at 19.85 °C (293 K).

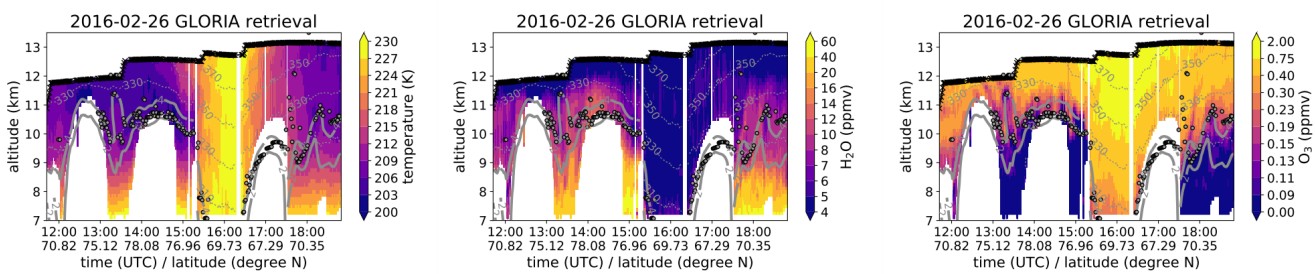

**Figure 10.** retrieval results derived from GLORIA measurements taken on 26th February 2016. The left panel shows temperature, the center panel shows water vapour, and the right panel shows ozone. Thick solid lines shows ECMWF potential vorticity, dotted lines shows potential temperature, and big dots show the location of the thermal tropopause.

## 4.3 Impact on retrieval

The impact of the uncertainties as derived from the PTB measurements above has been quantified for a measurement flight of one of our recent campaigns. Here, we compare the retrieval results, i.e., derived trace gases, against retrieval results gained from radiances perturbed by calibration uncertainties. Rather arbitrarily, flight 14 from the POLSTRACC campaign of winter

5 2015/16 was chosen, as this flight covers both situations with rather high and rather low stratospheric temperatures and thus measured radiances. This flight took place on 26th February 2016. The retrieval setup used is described by Ungermann et al. (2015). We focus here solely on the climate relevant variables of temperature, water vapor and ozone.

The derived fields are shown in Fig. 10. The results show rather typical stratospheric air masses for the time of the year.

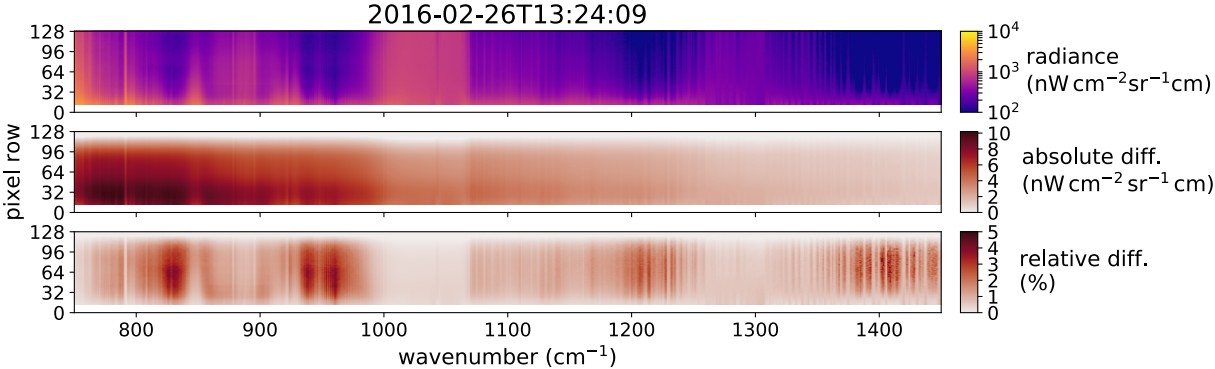

**Figure 11.** Radiances and perturbations for one image acquired at 13:24:09 (UTC). The top panel shows the measured radiances in a logarithmic color scale. The middle panel shows the absolute differences introduced by the blackbody inhomogeneity. The bottom panel shows the relative differences in percentages. The values for the lowermost pixel rows were already discarded during a previous processing step due to the presence of clouds.

The measured radiances are modified by applying Eq. (13), using the error fields of Sect. 4.2. Several variations of this error field as changing the sign or orientation of the field have been tested as well. The largest effect on the retrieval results is achieved by taking the measured deviation of one blackbody as it is, but changing the sign for the deviation of the other blackbody. The effect on one single measured radiance profile is depicted in Fig. 11. The perturbations exhibit a spectral and spatial variation, as the introduced deviation depends on the measured radiance itself. In general, the relative deviations are smaller for strong lines like the $CO_2$ Q-branch at $793\,cm^{-1}$ or the strong ozone lines around $1050\,cm^{-1}$ and larger for the atmospheric windows.

Continuing to the derived quantities, Fig. 12 shows the absolute and relative differences between the original and the perturbed retrieval results. For temperature a rather constant bias is introduced, which is in the order of the threshold value of Tab. 1. For the trace gases, the relative difference is in the order of the target thresholds, except for ozone at lower altitudes, where the volume mixing ratios are very low. Looking at the median, which is less affected by outliers, the relative difference stays in average below the target threshold of $1\,\%$. The results demonstrate that for the given inhomogeneity, the effect on the retrieval results stays in average below the thresholds and thus allows for long term studies of climate relevant variables.

## 5   Conclusions

This study has investigated the propagation of temperature deviations of the calibration blackbody through the GLORIA retrieval algorithm. A Monte Carlo approach has been used to derive uncertainty distributions of the retrieval results according to differently shaped deviations of the blackbody temperature. The shape and strength of the derived uncertainty distributions depend on the correlation length and standard deviation of the blackbody temperature inhomogeneity. In order to be able to

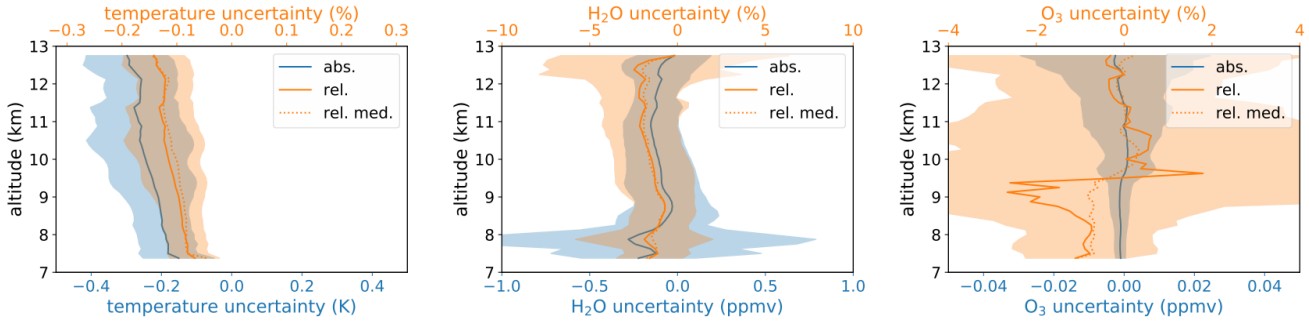

**Figure 12.** Absolute (blue) and relative (orange) differences between original and perturbed retrieval results for temperature (left), water vapor (center), and ozone (right). Solid lines show the mean differences over the whole flight, the dotted orange line shows the relative median, and the shaded areas show the standard deviation for the 274 evaluated profiles over the flight.

resolve typical trends in atmospheric species as temperature, ozone, and water vapor, the retrieval errors should remain below 20 % of the trend. To observe decadal trends, the resulting maximum tolerable temperature deviation of the blackbody temperature from the mean value for a typical correlation length of 10 vertical detector pixels is at 200 mK with a target of 100 mK. The target value is mostly reached for the blackbodies used for the calibration of GLORIA.

5  With the current data processing and retrieval setup, the temperature and emissivity uncertainties of the GLORIA blackbodies are among the leading sources of uncertainties, together with the uncertainty of the detector non-linearity determination and the instrument offset determination. Spectroscopic uncertainties and interfering species can have an impact of similar magnitude on the retrieval results. The latter sources of uncertainty can be reduced by further instrument characterization and an optimized retrieval setup, such that the uncertainties of the calibration blackbodies may become the leading error source.

10 *Competing interests.* The authors declare that they have no conflict of interest.

*Acknowledgements.* We thank Friedhelm Olschewski from BUW for the design of the GLORIA blackbodies, the successful improvement with the monolithic backplane and his contribution to the first calibration campaigns. The valuable support of Marco Schulz (PTB) during the radiometric characterization campain of the GLORIA reference blackbodies at PTB is gratefully acknowledged.

  We gratefully thank the GLORIA team of JÜLICH, and KIT, the POLSTRACC coordination team, and DLR-FX for operating the GLO-
15 RIA instrument and successfully conducting the POLSTRACC field campaign. The POLSTRACC campaign was supported by the German Research Foundation (Deutsche Forschungsgemeinschaft, DFG Priority Program SPP 1294).

  This work was funded by the European Metrology Research Program (EMRP), Project ENV53, 'Metrology for Earth Observation and Climate 2'. The EMRP is jointly funded by the EMRP participating countries within EURAMET and the European Union. This work was partly supported by the Bundesministerium für Bildung und Forschung (BMBF) under the project ROMIC/GW-LCYCLE, subproject 2

(01LG1206B) and subproject 3 (01LG1206C), as well as by the European Space Agency (ESA) under contract 4000115111/15/NL/FF/ah (GWEX).

We acknowledge support by the Deutsche Forschungsgemeinschaft and Open Access Publishing Fund of Karlsruhe Institute of Technology.

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
