# Peer review of "Characterization of blackbody inhomogeneity and its effect on the retrieval results of the GLORIA instrument"

_Atmospheric Measurement Techniques, 2018_

## Referee Comment (RC1) · F. Dupont (Referee) · 10 Apr 2018

The manuscript proposes a threshold for the maximum tolerable uncertainty of the blackbody temperature for O3, H2O and temperature profile for the GLORIA instrument. The study considers the viewing geometry of the optical system and derives the blackbody nonuniformities through the retrieval algorithms. The optical performance of the instrument is not considered (point spread function, cross-talk, etc), but I do not think that it would change the main results. This study is of great interest for the understanding of the technology and to improve the domain knowledge. The O3 requirement points to a value of 100 mK while the H2O would require larger values (less

stringent). I wish this study to be considered for any mission in order to adjust the hardware requirements accordingly. I have some minor comments listed below.

Page 9, line 4 : The text indicates a range from 10-100 while Figure 5 rather suggests 10-15 pixels. Please justify the 10-100 range.

Page 9, line 11 : The text associates the Âń temperature standard deviation of the blackbody Âż to the Âń blackbody uncertainties Âż and indicates that the O3 retrieval requires a 100 mK value. I would suggest to clarify that the 100 mK corresponds to (1-sigma) with the mention Âń 100 mK (1-sigma) Âż. You could also consider to refer to the expanded uncertainty of measurement for the blackbody properties. The expanded uncertainty of measurement is linked to the standard deviation by means of the coverage factor (http://www.european-accreditation.org/publication/ea-4-02-m-rev01–september-2013).

Page 13, Figure 8 : The figure on the right covers pixels 1 to 48 in the horizontal direction and pixels 1 to 128 in the vertical direction. The left figure covers the whole 128 x 128 pixels. I guess that the 1-48 area corresponds to the Horizontal Position 0-22.5mm, but I am not sure. Please clarify which area corresponds to whom in both figures.

---

## Referee Comment (RC2) · J. Taylor (Referee) · 30 Apr 2018

The manuscript presents a study that investigates the required uncertainty of the on-board radiometric reference standards to properly resolve decadal trends of retrieved water vapor, temperature profiles, and trace species for the Gimballed Limb Observer for Radiance Imaging of the Atmosphere (GLORIA). As noted in the manuscript, traceability to the SI is necessary for rigorous comparison of measurements between multiple instruments and is extremely important for decadal climate trending. The paper is well-written and represents a substantial scientific contribution within the scope of the journal.

[Figure]

I recommend that the paper be accepted subject to small revisions. My comments are included below.

General Comments:

The study focuses on the impact of temperature and emissivity uncertainties of the on-board radiometric reference standards and does not address other sources of uncertainty in the calibration or retrieval algorithms, which may or may not be more significant contributors to the overall uncertainty. While a comprehensive uncertainty analysis of all contributors is not the focus of this study, it would be helpful to briefly indicate if the onboard reference standard temperature and emissivity uncertainties are expected to be the primary source of uncertainty for the retrieved quantities and to note their relative significance with respect to the other identified sources of uncertainty (calibration and retrieval).

The uncertainty coverage factor (k) is only specified in one instance in the manuscript (page 10, line 11), and it is not clear if this is for the total uncertainty in the GLORIA blackbody temperature when using VIRST as a transfer standard and the VLTBB as a radiometric standard, or if it is the uncertainty of the of only the VIRST measurement, or the VLTBB as a source. "The typical uncertainty at -40C at a wavelength of 10 $\mu$m is 100 mK (k=2)." Given the identified importance of traceability to the SI, it would be useful to utilize expanded uncertainty notation throughout, with the coverage factor (k) explicitly noted when an uncertainty is specified. Alternatively, a brief note that all uncertainties are of a specified coverage factor or confidence unless otherwise noted would be sufficient.

Specific Comment: Section 4.1

Equation 18 and figure 6 assume that the effective emissivity is constant with wavenumber. Providing a statement regarding the expected spectral variability of Nextel 811-21 and/or uncertainty within the spectral range of the GLORIA measurement would be a useful clarification.

Specific Comment: Section 5

The conclusion would be further strengthened by explicitly noting what level of climate trends can be detected and over what time period, given the uncertainties in the GLORIA retrieved products discussed in the manuscript. Additionally, please consider including a summary statement in the conclusion that clearly states whether the total uncertainty in the retrieved products is expected to be driven by the temperature and emissivity uncertainties in the onboard radiometric references (the subject of this manuscript), or if other uncertainty contributors in the radiometric and spectral calibration or retrieval algorithms are expected to be the dominant uncertainties.

Editorial comment: Section 1, line 10

I believe that 'earth' should be capitalized in this context.

---

## Author Comment (AC1) · 30 May 2018

We thank the referee F. Dupont for reviewing the paper and the valuable comments. Our answers to the comments are given below. Relevant referee comments are inserted *in italics*.

*Page 9, line 4 : The text indicates a range from 10-100 while Figure 5 rather suggests 10-15 pixels. Please justify the 10-100 range.*

Figure 5 only demonstrates the general relation between the error in blackbody temperature (abscissa) and the error in the retrieved atmospheric parameter (color coded)

for different correlation lengths of the blackbody error as derived from the Monte-Carlo study. It does not give any information about the actual correlation lengths of the GLORIA blackbodies.

Our justification that our blackbody temperature error has correlation lengths of 10-100 pixels is based on the physical considerations that (a) due to the thermal conductivity of the material, temperature variations across the surface have to be rather smooth and (b) the instrument is focused to infinity and therefore one pixel "sees" a spot of about 30 mm diameter on the blackbody surface.

We have added a corresponding sentence on page 6, line 13, in order to make this more clear: "Due to the optical setup and the thermal conductivity of the blackbody surface, typical correlation lengths of the GLORIA blackbodies are expected to be in the range of 10 to 100 pixels. Temperature variations across the emitting surface have to be rather smooth because of the thermal conductivity of the material. Furthermore, the instrument is focused to infinity and the picture of the blackbody backplane on the detector is not sharp. Therefore correlation lengths below 10 pixels are not expected."

*Page 9, line 11 : The text associates the temperature standard deviation of the blackbody to the blackbody uncertainties and indicates that the O3 retrieval requires a 100 mK value. I would suggest to clarify that the 100 mK corresponds to (1-sigma) with the mention 100 mK (1-sigma)*

Following a suggestion from reviewer 2, we have now stated at the end of the introduction, that all uncertainties shall be understood as 1-sigma standard deviation values if not indicated otherwise. We also slightly changed the formulation on page 9, line 11, to make more clear that the values given are standard deviations.

Page 2, line 5, sentence added: "All uncertainties given in the text shall be understood as 1 $\sigma$ values if not indicated otherwise."

Page 9, line 11, sentence changed to: "Thus, for the GLORIA instrument and the applied retrieval setup, the ozone retrieval sets the threshold requirement of the standard deviation of the blackbody uncertainties to 200 mK and the desired target to 100 mK."

*Page 13, Figure 8 : The figure on the right covers pixels 1 to 48 in the horizontal direction and pixels 1 to 128 in the vertical direction. The left figure covers the whole 128 x 128 pixels. I guess that the 1-48 area corresponds to the Horizontal Position 0-22.5mm, but I am not sure. Please clarify which area corresponds to whom in both figures.*

The area covered by the detector array is approximately from -15 mm to +15 mm horizontally and from -40 mm to +40 mm vertically.

We have added this information in the text on page 12, line 3: "The area covered by the detector array is approximately from -15 mm to +15 mm horizontally and from -40 mm to +40 mm vertically."

---

## Author Comment (AC2) · 30 May 2018

We thank the referee J. Taylor for reviewing the paper and the valuable comments. Our answers to the comments are given below. Relevant referee comments are inserted *in italics*.

*General Comments: The study focuses on the impact of temperature and emissivity uncertainties of the on- board radiometric reference standards and does not address other sources of uncertainty in the calibration or retrieval algorithms, which may or may not be more significant contributors to the overall uncertainty. While a comprehensive uncertainty analysis of all contributors is not the focus of this study, it would be helpful*

[Figure]

*to briefly indicate if the onboard reference standard temperature and emissivity uncertainties are expected to be the primary source of uncertainty for the retrieved quantities and to note their relative significance with respect to the other identified sources of uncertainty (calibration and retrieval).*

The uncertainties of the calibration blackbody are comparable to other instrument uncertainties (e.g. detector non-linearity and instrument offset). Further relevant sources of uncertainty for the retrieval are the quality of the spectroscopic data and the impact of interfering species.

We have added an additional section at the end of section 2 (page 5, line 29) where these points are briefly discussed:

"2.4 Other sources of uncertainty

Uncertainties in the gain function do not only come from uncertainties of the temperature and emissivity of the blackbody but also from uncertainties in the determination of the detector non-linearity and the instrument self-emission (offset). These uncertainties are currently estimated to about 2 - 3 % (k=2). Further characterization is under way in order to reduce these uncertainties in future processing versions.

Beside the uncertainties in the calibrated spectra, the quality of the retrieval results also relies on the quality of the spectroscopic data and the impact of interfering species in the selected microwindows. Spectroscopic uncertainties are, however, systematic over time and cancel out in case of trend analyses. Furthermore, the quality of the spectroscopic data may improve in the future. Some interfering species impose uncertainties within the troposphere in the same order as the blackbody inhomogeniety. Further improvements on the retrieval are likely able to reduce these.

With the current data processing and retrieval setup, the blackbody temperature uncertainties are estimated to be one of the leading sources of uncertainty, together with other uncertainties in the gain determination and the impact of interfering species.

Since the other major sources of uncertainty are expected to be reduced with better instrument characterization and an optimized retrieval setup, the focus of this study is on the impact of the blackbody temperature uncertainties on the retrieval results relevant for the detection of climate relevant trends."

*The uncertainty coverage factor (k) is only specified in one instance in the manuscript (page 10, line 11), and it is not clear if this is for the total uncertainty in the GLORIA blackbody temperature when using VIRST as a transfer standard and the VLTBB as a radiometric standard, or if it is the uncertainty of the of only the VIRST measurement, or the VLTBB as a source. "The typical uncertainty at -40 °C at a wavelength of 10 µm is 100 mK (k=2)."*

This is indeed the total uncertainty in the GLORIA blackbody temperature when using VIRST as a transfer standard and the VLTBB as a radiometric standard. This is now stated more clearly on page 10, line 10:

"The typical uncertainty for the GLORIA blackbody surface radiation temperature at -40 °C at a wavelength of 10 µm is 100 mK (k=2) and is decreasing for higher temperatures.

*Given the identified importance of traceability to the SI, it would be useful to utilize expanded uncertainty notation throughout, with the coverage factor (k) explicitly noted when an uncertainty is specified. Alternatively, a brief note that all uncertainties are of a specified coverage factor or confidence unless otherwise noted would be sufficient.*

At the end of the introduction, we have included the general statement that all uncertainties given in the text shall be understood as $1\sigma$ values if not indicated otherwise.

Page 2, line 5, sentence added: "All uncertainties given in the text shall be understood as $1\sigma$ values if not indicated otherwise."

*Specific Comment: Section 4.1 Equation 18 and figure 6 assume that the effective emissivity is constant with wavenumber. Providing a statement regarding the expected spectral variability of Nextel 811-21 and/or uncertainty within the spectral range of the*

*GLORIA measurement would be a useful clarification.*

We have added a statement on the expected spectral variability of Nextel 811-21 in the relevant spectral range on page 10, line 30:

"This slope can only be real if either the wall coating has a significant slope or if the temperature non-uniformity within the measured area is of a similar magnitude as the slope, i.e. 200 mK. The coating Nextel 811-21 can be safely assumed to have a constant value of 0.97 in the spectral range from $500\,\mathrm{cm}^{-1}$ to $1500\,\mathrm{cm}^{-1}$ with only a slight spectral variability within a given uncertainty of about 0.01 but no significant overall spectral trend or slope (Adibekyan et al., 2017).

In order to exclude a temperature non-uniformity in the order of 200 mK, lateral scans were performed with VIRST ..."

*Specific Comment: Section 5 The conclusion would be further strengthened by explicitly noting what level of climate trends can be detected and over what time period, given the uncertainties in the GLORIA retrieved products discussed in the manuscript. Additionally, please consider including a summary statement in the conclusion that clearly states whether the total uncertainty in the retrieved products is expected to be driven by the temperature and emissivity uncertainties in the onboard radiometric references (the subject of this manuscript), or if other uncertainty contributors in the radiometric and spectral calibration or retrieval algorithms are expected to be the dominant uncertainties.*

We have changed the conclusions accordingly (starting page 15, line 7):

"In order to be able to resolve typical trends in atmospheric species as temperature, ozone, and water vapor, the retrieval errors should remain below 20 % of the trend. To observe decadal trends, the resulting maximum tolerable temperature deviation of the blackbody temperature from the mean value for a typical correlation length of 10 vertical detector pixels is at 200 mK with a target of 100 mK. The target value is mostly

reached for the blackbodies used for the calibration of GLORIA.

With the current data processing and retrieval setup, the temperature and emissivity uncertainties of the GLORIA blackbodies are among the leading sources of uncertainties, together with the uncertainty of the detector non-linearity determination and the instrument offset determination. Spectroscopic uncertainties and interfering species can have an impact of similar magnitude on the retrieval results. The latter sources of uncertainty can be reduced by further instrument characterization and an optimized retrieval setup, such that the uncertainties of the calibration blackbodies may become the leading error source."

*Editorial comment: Section 1, line 10 I believe that 'earth' should be capitalized in this context*

Yes, corrected.